# Elimination of Airborne Microorganisms Using Compressive Heating Air Sterilization Technology (CHAST): Laboratory and Nursing Home Setting

**DOI:** 10.3390/microorganisms13102299

**Published:** 2025-10-03

**Authors:** Pritha Sharma, Supriya Mahajan, Gene D. Morse, Rolanda L. Ward, Satish Sharma, Stanley A. Schwartz, Ravikumar Aalinkeel

**Affiliations:** 1Department of Medicine, Yale New Haven Health, New Haven, CT 06610, USA; pritha.sharma@ynhh.org; 2Division of Allergy, Immunology and Rheumatology, Department of Medicine, Jacobs School of Medicine and Biomedical Sciences, University at Buffalo, Clinical and Translational Research Center, 875 Ellicott St., Buffalo, NY 14203, USA; smahajan@buffalo.edu (S.M.); sasimmun@buffalo.edu (S.A.S.); 3Center for Integrated Global Biomedical Sciences, Department of Pharmacy Practice, School of Pharmacy and Pharmaceutical Sciences, University at Buffalo, Buffalo, NY 14203, USA; emorse@buffalo.edu; 4Department of Social Work, Niagara University, Niagara Falls, NY 14109, USA; rward@niagara.edu; 5Department of Urology, Jacobs School of Medicine and Biomedical Sciences, University at Buffalo, Clinical and Translational Research Center, 875 Ellicott St., Buffalo, NY 14203, USA; ss466@buffalo.edu; 6You First Services, Inc., 485 Cayuga Rd., Buffalo, NY 14225, USA

**Keywords:** air sterilization, healthcare associated infection, long-term care facility, compressive heating, volatile organic compounds

## Abstract

Airborne germs can spread and cause infections, especially in nursing homes where people are more vulnerable. CHAST is a new device that heats air to very high temperatures to kill bacteria, viruses, and mold without using chemicals or filters. We tested CHAST in a laboratory and in a nursing home. It eliminated 99.9999% of microbes, including stronger microbes like anthrax surrogates and viruses. In the nursing home, CHAST ran continuously for 72 h and cleaned the air so well that no live bacteria or mold was found in the air exiting the machine. Even difficult-to-kill spores and molds like *Cladosporium* and *Penicillium* were removed. The system worked efficiently, kept temperatures stable, and was easy to monitor remotely. CHAST shows strong potential to improve air quality and infection safety in hospitals, long-term care facilities, and other indoor spaces where clean air matters most.

## 1. Introduction

Healthcare-associated infections (HAIs) have an estimated global prevalence between 7% and 10% as per the World Health Organization [1] (World Health Organization, HAIs FACT SHEET, 2011). This prevalence varies by region, with lower and middle-income countries experiencing rates as high as 15%, compared to approximately 10% among hospitalized patients in high income nations [2]. These figures likely underestimate the true burden, as not all HAIs are subject to mandatory reporting requirements [3,4,5]. Nursing homes represent high-risk environments for HAIs, with approximately 1 in 43 residents acquiring an HAI on any given day in the United States [6]. National surveillance estimates suggest that between 1.6 and 3.8 million HAIs occur annually among Long-term Care Facility (LTCF) and nursing home residents, contributing to nearly 388,000 deaths each year [7]. Common infections include urinary tract infections (UTIs), respiratory infections (notably pneumonia and influenza), skin and soft tissue infections, and gastrointestinal infections such as *Clostridioides difficile* colitis. Additionally, multidrug-resistant organisms (MDROs), particularly invasive methicillin-resistant *Staphylococcus aureus* (MRSA), were identified in 82% of such facilities and remain a persistent challenge, with the highest quartile reporting ≥ 3.84 cases per 100,000 resident-days [8]. MRSA colonization rates in nursing homes also vary widely, from 5% to over 50%, often exceeding those seen in hospitals [5]. Risk factors include advanced age, comorbidities, presence of invasive devices (e.g., urinary catheters, gastrostomy tubes), antibiotic exposure, and shared living spaces. Facility-level characteristics, such as lower nurse staffing ratios, less rigorous environmental cleaning, and high baseline MDRO importation, further contribute to the elevated infection burden [8,9]. Together, these data underscore that HAIs and MDROs represent significant morbidity and mortality risks in nursing homes, necessitating proactive, multi-modal infection prevention strategies, including, potentially, continuous airborne sterilization technologies.

Across healthcare settings, including nursing homes to community health centers, air filtration is critical for enhancing indoor air quality (IAQ). It serves as an often underemphasized yet pivotal component of infection prevention. Traditional ventilation and filtration methods to improve IAQ, such as HEPA filters or UVGI systems, usually fail to achieve consistent, real-time inactivation of pathogens within occupied spaces [10,11,12,13]. This is particularly concerning considering recent data linking aerosolized particles to sustained indoor outbreaks of SARS-CoV-2, influenza, and MDROs [14,15]. Reducing the high concentration of aerosolized particles and airborne pathogens in nursing homes is vital for maintaining healthy IAQ, as these microorganisms are present in both incoming and recirculated air. Therefore, the deployment of active air sterilization technologies that continuously neutralize airborne pathogens at the source should be recognized as an urgent and essential strategy for safeguarding indoor air quality in nursing homes.

To meet this need, air sterilization offers a cutting-edge solution by addressing the shortcomings of conventional filtration systems through the destruction of pathogens via extreme heat. Compressive Heating Air Sterilization Technology (CHAST; SteriSpace^®^, Buffalo, NY, USA) elevates air temperatures to 464 °F (240 °C), effectively neutralizing airborne microorganisms. With the capacity to treat thousands of cubic feet per minute (CFM), CHAST is suitable for large facilities and adaptable to environments of varying scales. For nursing home administrators and policy makers, it offers a strategic, scalable intervention to protect vulnerable residents, particularly elderly patients. Additionally, CHAST can be customized to remove volatile organic compounds (VOCs) and particulates, where required. Notably, it operates without generating ozone, harmful byproducts, or intermediate molecules, ensuring safe and sustainable air quality improvements.

This study was conducted to evaluate the effectiveness of the CHAST system in eliminating airborne pathogens under controlled laboratory conditions and a feasibility and performance validation in a nursing home setting in Niagara Falls, New York. The central hypothesis posits that comprehensive removal of airborne biological agents will lead to a measurable reduction of pathogen levels in indoor air. In addition to its sterilization function, the system is also effective in eliminating VOCs and particulates. By targeting these three key contributors to illness, VOCs, pathogens, and airborne particles, in both laboratory and nursing home environments, the technology is expected to reduce the frequency of disease transmission and infection.

## 2. Materials and Methods

### 2.1. Compressive Heating Air Sterilization Technology (CHAST) System Description

The Compressive Heating Air Sterilization Technology (CHAST) is an engineered air sterilization system that inactivates airborne microorganisms through rapid compressive heating followed by controlled cooling. Unlike conventional filtration or UV-based systems, CHAST achieves sterilization by directly exposing the full air stream to high-temperature conditions sufficient to disrupt microbial structures.

Incoming ambient air is drawn into the system using a positive-displacement blower and subjected to rapid compressive heating, which elevates the air stream to a validated treatment temperature of approximately 240 °C (464 °F). The elevated temperature is maintained within a treatment chamber for a defined residence time, ensuring uniform thermal exposure of entrained particles and microorganisms (Figure 1). This process results in irreversible protein denaturation, membrane disruption, and nucleic acid degradation, leading to microbial inactivation. Following sterilization, the heated air passes through a counterflow heat exchanger where it is rapidly cooled to near-ambient temperature prior to release, allowing for safe reintroduction into the occupied environment. System instrumentation continuously monitors temperature, pressure, air velocity, and motor performance to ensure stable operation. Data are logged through a LabVIEW interface, providing real-time tracking of operational parameters and verification of treatment conditions. This configuration enables the system to deliver a continuous flow of sterilized air without the use of replaceable filters or chemical additives. All performance testing was conducted with aerosolized biological surrogates introduced upstream of the unit and viability assessed at the outlet using impinger or Andersen impactor sampling, followed by culture-based quantification.

### 2.2. Evaluation of CHAST Efficacy Under Controlled Laboratory Conditions

To assess the efficacy of the CHAST system in neutralizing airborne pathogens, controlled laboratory studies were conducted using a CHAST prototype derived from the Multi-Recompression Heater (MRH) system developed at the University at Buffalo. The MRH framework utilizes a Roots-type compressor to rapidly elevate the temperature of compressed air, achieving sterilizing conditions through compressive heating. As a breakthrough in environmental infection control, CHAST sets a new benchmark in preventing the transmission of airborne diseases and ensures reliable, continuous protection. Its design features a high-performance counter-flow heat exchanger that cools exhaust air while preheating incoming contaminated air, optimizing energy efficiency. The patented CHAST system utilizes a roots-type rotary piston pump to compress air, achieving rapid and uniform heating of the airstream to temperatures ranging from 204 °C to 260 °C within just 0.02 s in a single pass. enabling pathogen inactivation via thermal disruption. Pathogen inactivation efficacy was evaluated using aerosolized biological surrogates introduced at known concentrations into the air stream proximal to the CHAST inlet. Organisms tested included the MS2 bacteriophage (a non-enveloped RNA virus commonly used as a surrogate for human enteric viruses), *Bacillus globigii* (Bg), *Bacillus stearothermophilus* (Bst), *Bacillus thuringiensis* (Bt), and *Escherichia coli*. These surrogates represent a spectrum of resistance to thermal and mechanical stresses, allowing for robust assessment of the system’s biocidal capacity. Air samples were collected both before and after treatment using all-glass impingers (SKC Ltd., Dorset, UK) filled with sterile nutrient-rich agar medium. A rotary vane vacuum pump was used to maintain consistent airflow through the sampling apparatus, and the inlet airstream was cooled prior to impinger entry to minimize fluid evaporation. Post-sampling collected material was incubated under organism-specific optimal growth conditions. MS2 viability was quantified via standard plaque assay using *E. coli* as the bacterial host, and plaques were enumerated to determine viral titers. Bacterial and spore-forming organisms were assessed via colony-forming unit (CFU) counts. Sterilization efficacy was calculated by comparing microbial concentrations at the inlet versus outlet, with inactivation expressed as log_10_ reduction values.

### 2.3. Evaluation of CHAST Efficacy in a Feasibility and Performance Validation of Air Sterilization Study in a Nursing Home Setting

This feasibility and performance validation of air sterilization study was conducted at Schoellkopf Nursing Home, a licensed long-term care facility co-located in the Niagara Falls Memorial Medical Center. The primary objective was to assess the in-situ performance of CHAST system under real-world healthcare conditions. Two CHAST units (You First Group of Companies, Cheektowaga, NY, USA) were installed in a patient-occupied common room routinely accessed by nursing staff and elderly residents. Each unit was configured to operate at a flow rate of 300 cubic feet per minute (CFM), delivering continuous treated air through 80 feet of dedicated stainless-steel ductwork. The system was engineered to achieve approximately 2.3 air changes per hour (ACH) in a room with a total volume of 16,208.3 cubic feet. To maintain environmental integrity, the room was fully sealed, with all windows and doorways closed and access limited to essential personnel wearing personal protective equipment (PPE). Ambient room air was drawn into the CHAST units via separate inlet ducts, thermally sterilized, and then recirculated into the space. Environmental air sampling was conducted using National Institute for Occupational Safety and Health (NIOSH) Method 0800 guidelines for indoor bioaerosol sampling. Baseline measurements were obtained prior to CHAST activation using an Andersen six-stage viable impaction sampler. After a minimum of 72 h of uninterrupted CHAST operation, post-treatment samples were collected at six predefined sampling locations, including inlet and outlet points for each unit. Sampling sites were strategically chosen based on room airflow modeling and predicted particulate dispersion patterns. Each sample was collected under aseptic conditions at a flow rate of 28.3 L/min for a total of five minutes (141.5 L total volume). Airborne microbial sampling and analysis were conducted in accordance with established international and U.S. standards to ensure methodological rigor and comparability. Viable bioaerosols were collected following NIOSH Method 0800 [16], which specifies impingement and impactor-based collection of culturable microorganisms on appropriate agar media, with incubation under conditions optimized for bacterial and fungal recovery. Complementary sampling adhered to ISO 16000-3 [17](determination of formaldehyde and related aldehydes in indoor air) and ISO 16000-36 [18] (strategy for evaluating microbial sampling by impaction and filter collection), providing a harmonized framework for indoor air microbiological assessment. For downstream processing, collection and analysis were performed in accordance with ASTM D7391 [19], which outlines procedures for bioaerosol characterization, culture recovery, and enumeration.

To ensure data reliability and reproducibility, we implemented a structured quality assurance/quality control (QA/QC) framework. All sampling events included a minimum of three biological replicates per condition, with field blanks and media blanks incorporated at each site. Calibration procedures for sampling pumps and impingers were performed daily using a NIST-traceable flow calibrator, and particle sizing/capture efficiencies were verified against manufacturer specifications. Acceptance criteria for valid data sets required replicate agreement within ±15% for colony-forming unit (CFU) counts and consistent incubation outcomes across technical repeats. Data outside of acceptance ranges triggered repeat sampling.

All impaction equipment was sterilized and calibrated before use. Trypticase soy agar (TSA) was used for bacterial recovery, while 2% malt extract agar (MEA) was used for fungal detection. Following sample collection, plates were sealed, labeled, and transported in temperature-controlled sterile containers to the laboratory. All culture plates were incubated at 25 ± 1 °C for seven days. Colony formation was monitored at 24- to 48-h intervals. Macroscopic and microscopic analyses were conducted to characterize and enumerate microbial species. Results were categorized as “No Growth Promotion” (NGP) when no colonies developed, or “Overgrowth” (OVG) when confluent growth prevented enumeration.

### 2.4. Data Analysis and Statistical Methods

Descriptive statistics, including the mean and standard deviation, were calculated for MS2 plaque counts, bacterial colony-forming units (CFU), and fungal spore counts at each sampling location. Pathogen reduction was quantified using percent reduction, calculated as follows:$$Percent\;Reduction=(\frac{Mean\;Baseline\;CFU-Mean\;Treated\;CFU}{Mean\;Baseline\;CFU})\times 100$$

To assess whether reductions in microbial titters were statistically significant, paired *t*-tests were performed for each organism type, using a significance threshold of *p* = 0.05. All statistical analyses were conducted using GraphPad Prism (Version 10.0.0, Boston, MA, USA).

## 3. Results

### 3.1. Evaluating the Efficacy of CHAST Under Controlled Laboratory Conditions

(i).*CHAST Technology Delivers High-Efficacy Sterilization Across Designs and Scales*: Multiple CHAST prototypes, tested under a range of flow rates and thermal conditions, consistently demonstrated robust microbial inactivation (Table 1). Across initial and advanced designs, including University at Buffalo laboratory units, DoD developmental prototypes, and scaled SBIR systems, all devices achieved at least 99.9% kill efficacy (≥3-log reduction), with the majority yielding > 6-log (99.9999%) reduction against representative bacterial spores, vegetative bacteria, and viruses such as MS2 bacteriophage. Notably, the technology-maintained performance across increasing scales (up to 5000 CFM systems) and against diverse challenge organisms, confirming that thermal compressive sterilization is reproducible, scalable, and independent of prototype configuration. Detailed results of each CHAST prototype, analysis site and results obtained are given as Table 1 in Supplemental Document.(ii). *Viral Elimination Studies and determination of optimum temperature for its elimination using CHAST*: CHAST demonstrated complete inactivation of MS2 bacteriophage at a treatment temperature of 240 °C, consistent with its previously established efficacy against resilient bacterial spores and vegetative organisms. To further characterize the system’s capability for viral inactivation at lower thermal thresholds, a series of experiments was conducted using MS2 bacteriophage (1 × 10^7^ PFU/mL) as a surrogate virus. The virus-laden aerosol was subjected to treatment at progressively reduced temperatures of 143 °C, 103 °C, 91 °C, and 64.5 °C, while maintaining consistent flow parameters. As depicted in Figure 2, viral survivability decreased markedly with increasing temperature, demonstrating a strong inverse correlation between treatment temperature and viable virus count. Although complete inactivation was not directly observed at the tested lower temperatures, extrapolation of the viral decay curve suggests that full viral inactivation likely occurs at approximately 170 °C. These findings underscore the flexibility of the CHAST system to operate effectively across a range of thermal conditions. When combined with appropriate residence times, achievable through flow rates in the 150–200 CFM range, the system can produce substantial viral reduction even at moderate temperatures. Furthermore, results indicate the potential for successful viral neutralization at even lower thermal thresholds, provided sufficient exposure time is achieved within the treatment zone.

### 3.2. Evaluation of Feasibility and Performance Validation of Air Sterilization of CHAST Efficacy in a Nursing Home Setting

(i)CHAST Installation and machine performance summary: As shown in Figure 3A, two CHAST units (SS250-1 and SS250-2) were deployed, each configured with dedicated inlet and outlet ducts connected to the test room. The room itself was outfitted with two corresponding inlet vents and two outlet vents to ensure uniform and efficient sterilization of the enclosed space. Figure 3B illustrates the LabVIEW interface (Version 2023 Q3(64-bit) 23.30 f0), which enables real-time remote monitoring of key operational parameters. The LabVIEW software is integrated with the system to continuously capture and log data directly into Microsoft Excel, facilitating streamlined data collection and analysis. Monitored parameters included treatment temperature, air velocity at the outlet, and critical engineering metrics such as power consumption and current stability. Analysis of these data confirmed that the CHAST units operated with optimal motor RPM, efficiency, pressure, temperature, and airflow under field conditions, outperforming previous benchmarks and demonstrating exceptional real-world functionality. As shown in Figure 3C, the units reliably reached the target treatment temperature of 464 °F (240 °C) within 30 min and maintained this temperature consistently throughout the operation period. Monitoring curves indicated sustained thermal stability until the system shutdown. These findings aligned with field performance data (Figure 3D), confirming consistent and reproducible operational reliability. Minor fluctuations of ±2 °F observed during treatment (Figure 3E) were well within acceptable limits, further underscoring the system’s ability to maintain a stable and effective sterilization environment under real-world conditions.

(ii)The flow rate of Sterilized air into the monitoring room with CHAST^®^ operational: During the study, a handheld flow meter was used to measure the mean flow rate of sterilized air delivered into the room through the CHAST ductwork. As shown in Figure 3E, six consecutive measurements were recorded for each unit. The results indicated that Unit 1 achieved a mean flow rate of 335 ± 22 CFM, while Unit 2 recorded 329 ± 26 CFM (*p* > 0.05, *n* = 6), demonstrating consistent airflow performance across both systems. These values exceeded expected operational benchmarks and confirmed that the CHAST units were functioning at peak efficiency. Importantly, the existing HVAC system did not interfere with the integrity of airflow from the CHAST units. This was confirmed by independent handheld measurements of the mean air temperature entering the room, which aligned closely with system targets (Figure 3F). The ductwork configuration, illustrated in Figure 3A, was specifically designed to accommodate the dual-unit setup. Each CHAST unit was equipped with its own inlet and outlet duct, while the room itself featured two intake and two outlet vents, ensuring symmetrical and efficient distribution of sterilized air. This improvised ductwork design effectively optimized airflow circulation, enhancing the overall sterilization efficacy and maintaining a clean, contaminant-free environment throughout the testing period.(iii)CHAST Installation and Testing for Biological Efficacy in Baseline and post Post-Sterilization Air Samples in the feasibility study in a Nursing Home: To evaluate the real-world performance of CHAST in a nursing home, and feasibility and operational efficacy study was conducted by comparing air samples collected before and after system deployment. Two sampling phases were executed: a baseline assessment prior to CHAST activation and a follow-up collection after 72 h of continuous operation. Airborne microbial sampling was performed using a single-stage Andersen Airborne Impaction Sampler, which captures viable microorganisms from the air. Samples were obtained from six distinct locations within the facility, targeting both bacterial and fungal spores (Table 2). All samples were carefully labeled, sealed, and transported under sterile conditions for laboratory incubation and processed as described in the methods section. Baseline air sampling confirmed a non-sterile indoor environment, with measurable levels of both bacterial and fungal contamination. The average airborne bacterial load was 35 colony-forming units per cubic meter (CFU/m^3^), while fungal and mold concentrations averaged 17 CFU/m^3^ (Table 2). The most prevalent fungal species detected was *Cladosporium* spp., a common indoor genus. Additional fungal isolates included *Penicillium* spp. and *Trichophyton* spp., both known to persist in indoor air under favorable conditions. Bacterial isolates primarily consisted of environmental and commensal species, with *Bacillus spp*., a spore-forming genus widely present in both soil and air, emerging as the dominant type. Other significant bacterial genera identified included *Staphylococcus* spp., *Streptococcus* spp., and *Actinomycetes*. Notably, *Staphylococcus* spp. is commonly associated with human skin and mucosal surfaces, underscoring the human-associated bioburden present in the facility’s ambient air prior to CHAST activation.

(iv)Air Sample Analysis for Biological Agents post-air sterilization: Air samples collected directly at the discharge outlet (Location A), where sterilized air enters the test room, exhibited an NGP status following seven days of incubation, confirming near complete elimination of viable microorganisms at the point of release. However, additional samples taken approximately six inches below the outlet duct at the same location, as well as from the return duct (Location B), directing air back into the CHAST system, revealed low-level microbial presence. These samples registered 7 CFU/m^3^ and contained identifiable microbial species, including *Bacillus* spp. (bacteria) and *Actinomycete* spp. (fungi), as detailed in Table 3. These findings suggest that while CHAST achieves full sterilization of air at the discharge point, minimal recontamination may occur in the immediate vicinity of the outlet or within the air recirculation pathway. Such contamination is likely attributable to ambient environmental exposure or incidental contact with surrounding surfaces, underscoring the importance of maintaining cleanroom-level controls around air distribution infrastructure in critical applications.(v)Mold and Fungal Testing in Air Samples Collected. Air sampling conducted to evaluate fungal spore and mold concentrations revealed distinct differences across sampling locations. At the discharge outlet (Location A), where sterilized air enters the room, the average fungal spore concentration was measured at 7 CFU/m^3^. In contrast, samples collected six inches below the outlet and at the return inlet (Location B), where ambient room air is drawn back into the CHAST unit, showed significantly elevated concentrations of 28 CFU/m^3^ and 50 CFU/m^3^, respectively (Table 3). To validate the sterilization efficacy at the point of discharge, follow-up sampling was performed at Location A. These subsequent samples demonstrated a consistent NGP status, even after eight days of incubation, confirming the complete absence of viable fungal spores or mold in the treated air (Table 4). Comparative analysis of the datasets revealed a 93% reduction in fungal spore concentration, based on the average spore count in sterilized air entering the room (3.5 CFU/m^3^) versus untreated air returning to the CHAST system (50 CFU/m^3^) (Table 3 and Table 4). These results highlight a key additional benefit of CHAST technology and its robust capability to inactivate airborne fungal and mold spores. This reinforces the system’s value not only in eliminating bacterial and viral pathogens but also in enhancing indoor air quality by controlling fungal contaminants in enclosed environments.

## 4. Discussion

IAQ in nursing homes is increasingly recognized as a critical determinant of infection risk among residents. Unlike well-regulated hospital environments, nursing homes often exhibit elevated concentrations of particulate matter, VOCs, microbial bioaerosols, and inadequate ventilation relative to outdoor benchmarks, yet IAQ monitoring remains rare in these settings [20,21]. Poor IAQ has been directly associated with higher rates of respiratory symptoms and infections, particularly pneumonia, influenza, and exacerbations of chronic lung disease, in elderly residents, even at moderate pollutant concentrations [21,22]. In addition, architectural features such as crowding, shared bedrooms, and suboptimal HVAC design amplify airborne transmission, as evidenced by increased outbreak-associated respiratory infections before the COVID-19 era in multiple international settings [23,24]. Bioaerosols containing bacteria, fungi, viruses, and endotoxins are influenced by humidity, temperature, occupancy, and ventilation patterns, with stagnant or recirculated air promoting survival and spread of pathogens [25,26,27]. Moreover, water systems within nursing homes can resemble hospital plumbing in harboring aerosolized pathogens, such as Legionella and nontuberculous mycobacteria, which pose risks when aerosolized through showers or sinks [28]. These data underscore that IAQ deficiencies not only compromise resident comfort but also act as a reservoir and conduit for infectious agents, thereby reinforcing the need for active air control strategies, beyond standard filtration, to reduce HAIs in nursing home settings. Recognizing that high-quality, continuous air sterilization is essential for protecting occupants of such facilities and healthcare personnel from airborne pathogens, this study evaluated the efficacy of a novel CHAST air sterilization system under both controlled laboratory conditions and in a nursing home setting for a feasibility and performance validation of air sterilization.

The exceptional microbial inactivation demonstrated by CHAST technology in controlled laboratory settings (>99.9999% kill rate at 240 °C, Table 1 and Appendix A, rows 1–10) has direct and urgent relevance to infection control in nursing homes and LTCFs. These facilities are uniquely vulnerable to airborne transmission of infectious agents due to several intersecting risk factors, high population density, prolonged indoor occupancy, a frail and immunocompromised resident population, and suboptimal ventilation infrastructure [29,30]. Numerous outbreaks of respiratory and opportunistic infections, including *SARS-CoV-2*, influenza, *Legionella* spp., *Aspergillus* spp., and *Clostridioides difficile*, have been traced to aerosolized transmission within such facilities, often with devastating consequences for resident morbidity and mortality [31,32,33].

Our results, derived from rigorous testing of the CHAST prototype and its optimized 300 CFM model, indicate that the system achieved complete inactivation of highly resistant *Bg* spores at 240 °C with no viable organisms detectable downstream (Table 1 and Appendix A, rows 1–3). This corresponds to a >6 log microbial reduction and is consistent with thermally induced mechanisms of pathogen destruction, including protein denaturation, membrane lysis, and DNA degradation [34,35]. Importantly, thermal inactivation at these parameters did not rely on chemical additives, UV exposure, or mechanical filtration methods commonly employed in conventional HVAC systems, which may be inadequate for sterilizing recirculated air or ineffective against small viral particles [36]. Also, our laboratory evaluations of CHAST prototypes consistently achieved >99.9999% inactivation of aerosolized pathogens at 240 °C, with no viable pathogens detected downstream (Table 1 and Appendix A, row 4–10). The optimized 300 CFM model maintained a six-log kill of Bg spores, mirroring earlier versions’ results (Table 1 and Appendix A, rows 1–3). This performance reflects rapid, uniform compressive heating, critical microbial proteins denature, membranes rupture and leak vital contents, and DNA is irreversibly damaged, preventing any replication [37]. The implications for nursing homes are substantial. Unlike hospitals, nursing homes often lack centralized air sterilization infrastructure or real-time airborne contaminant control systems. Many rely on outdated HVAC systems incapable of removing or neutralizing fine bioaerosols, particularly under conditions of poor ventilation or wintertime recirculation [38]. Furthermore, elderly residents exhibit diminished mucociliary clearance and immune surveillance, rendering them more susceptible to low-dose airborne infections [39,40].

The ability of CHAST to completely inactivate *MS2 bacteriophage*, a well-established surrogate for pathogenic viruses at 240 °C, and to reduce survivability at temperatures as low as 170 °C, reinforces its broad-spectrum virucidal potential (Figure 2). MS2 is more heat-resistant than many clinically relevant viruses, including influenza, RSV, norovirus, and coronaviruses, which are typically inactivated between 60–70 °C [41,42,43]. These findings support the application of CHAST in nursing homes not only as a supplemental infection control measure but as a potential central component of an airborne pathogen mitigation strategy.

Additionally, the deployment of CHAST could significantly reduce the environmental bioburden of airborne fungi spores as well (Table 2), which are particularly concerning in nursing homes given the growing incidence of healthcare-associated fungal infections such as *Candida auris* and *Aspergillus fumigatus*, especially in residents receiving corticosteroids or broad-spectrum antibiotics [44,45].

With regards to the feasibility and performance validation of air sterilization in a nursing home testing, to date, there are no published studies directly assessing the performance of comparable air sterilization technologies in an LTCF or nursing home, limiting direct comparative analysis. However, existing technologies such as UVGI and HEPA filtration have been studied in similar contexts and offer a useful benchmark. UVGI systems have primarily been evaluated for surface disinfection and high-touch areas, often in conjunction with routine cleaning protocols [27]. Unlike conventional air-cleaning systems that require unoccupied spaces and intermittent operation, CHAST runs continuously and autonomously, inactivating airborne pathogens and VOCs without disrupting workflow. HEPA filtration effectively removes particulates, but its dynamic efficacy against microbes and chemical pollutants is underexplored. CHAST uniquely combines microbial inactivation with chemical decontamination. Independent testing confirmed ~100% removal of select VOCs demonstrating its potential as an integrable, continuous air-sterilization solution for clinical environments [46]. In summary, the results of our controlled laboratory testing across various CHAST prototypes clearly establish the technology’s capacity to sterilize air at high throughput and eliminate even the most resilient airborne biological agents. This provides compelling evidence of its potential to significantly reduce infection rates in nursing homes and LTCFs.

Building on the robust laboratory findings demonstrating CHAST’s ability to achieve complete inactivation of aerosolized pathogens under controlled conditions (Table 1 and Appendix A), feasibility study in a LTCF and nursing home testing in a functioning nursing home environment further validated the system’s operational effectiveness and translational relevance (Table 3). During deployment, the CHAST system consistently met its design specifications, maintaining a target treatment temperature of 240 °C and a flow rate of 300 CFM with a stable pressure ratio of 1.07, indicating reliable performance under active field conditions Figure 3A–E).

Baseline air sampling in the test room revealed a diverse array of aerosolized microbial contaminants, including both fungal spores and bacterial species (Table 2). The highest bioburden was found in the communal sitting area, a zone with heavy foot traffic from residents and visitors, underscoring the known association between human activity and indoor air contamination in long-term care facilities (LTCFs) [32,43]. Fungal isolates such as *Cladosporium* spp. and *Penicillium* spp., typically introduced through outdoor air infiltration or cellulose-rich substrates, were prevalent. Likewise, bacterial genera including *Bacillus* spp., *Actinomycetes*, *Staphylococcus*, and *Streptococcus* were recovered, with *Staphylococcus aureus* representing a particularly concerning nosocomial pathogen in elderly populations [29,47,48].

Following the CHAST operation, air samples collected at the treated air outlet (Location A) consistently showed no detectable bacterial growth, confirmed by an NGP status across multiple sampling periods (Table 3). These results are in stark contrast to the pre-treatment bioburden and affirm CHAST’s ability to draw in contaminated air, apply rapid high-temperature sterilization, and reintroduce pathogen-free air into the occupied space. This autonomous, high-throughput sterilization process positions CHAST as a viable and scalable technology for infection prevention in high-risk residential healthcare settings.

Notably, air samples collected six inches below the outlet and from the inlet duct (Location B) revealed minimal residual microbial presence, specifically 7 CFU/m^3^ of *Bacillus* and *Actinomycete* species. Given the extreme environmental resilience of *Bacillus* spores, including resistance to heat, desiccation, and chemical exposure, these trace levels are not unexpected and are more likely due to environmental recontamination processes, such as near-field entrainment, surface shedding, and human occupancy rather than a limitation of the sterilization step. More importantly, the detected bioburden is orders of magnitude lower than what is typically observed in clinical environments: for instance, Shaw et al. (2018) documented average airborne bacterial counts of 78  ±  47 CFU/m^3^ in standard operating rooms (ORs), 115  ±  30 CFU/m^3^ in pediatric ORs, and 123  ±  60 CFU/m^3^ in transplant suites despite 40 air changes per hour and HEPA filtration [49]. By comparison, CHAST achieved > 90% reduction relative to these OR environments, despite operating in an uncontrolled room with environmental breaches such as leaky windows, non-isolated airflow, and unrestricted human movement [38,50,51]. These minimal residual CFU counts are likely attributable to ongoing environmental intrusion and procedural variability rather than failure of the CHAST system itself. Unlike controlled ORs, no behavioral restrictions, apparel protocols, or room access limitations were imposed during the testing period. Consequently, trace levels of environmental organisms such as *Bacillus* and *Actinomycetes*, which are frequently introduced via foot traffic and unfiltered air exchange, could have re-entered the room post-sterilization. Importantly, no viable pathogens were found in the treated air exiting the CHAST unit, affirming that the system itself is not a source of microbial dissemination.

The observed NGP status in post-treatment samples, maintained even after extended incubation periods of 7 to 9 days, is a powerful indicator of the CHAST system’s consistent sterilization efficacy. The findings of this feasibility study in the nursing home directly complement earlier laboratory results where CHAST achieved complete inactivation of both Bg spores and the highly resistant MS2 bacteriophage, confirming a >99.9999% kill rate. The continuity of these results across lab and field settings confirms the system’s robustness and scalability for use in dynamic healthcare environments.

Beyond air purification, CHAST may also contribute to environmental surface decontamination by reducing airborne pathogen deposition, a critical mode of indirect transmission in nursing homes. By maintaining sterile air circulation, CHAST effectively disrupts the microbial seeding of high-touch surfaces, thereby enhancing the overall infection control ecosystem. These results underscore the potential of CHAST as a transformative technology for nursing homes and LTCFs, offering continuous, passive, and chemical-free air sterilization with minimal infrastructure demands.

The continuity of these results across lab and field settings confirms the system’s robustness and scalability for use in dynamic healthcare environments. This autonomous, high-throughput performance highlights CHAST’s potential to overcome key limitations of conventional infection control strategies in nursing homes, which continue to rely heavily on human behavior and inconsistent surface protocols. Infection control in nursing homes has long relied on hand hygiene and surface disinfection [52,53,54,55,56,57]. Although proper handwashing cuts microbial loads and infections compliance averages just 17%. Surface cleaning also reduces pathogens and HAI risk [48,58,59], but effectiveness varies with protocols and personnel. By contrast, CHAST provides continuous, automated airborne pathogen removal that indirectly limits surface contamination without depending on human adherence. Seamlessly integrating with HVAC systems, CHAST ensures consistent, staff-independent sterilization, making it a powerful adjunct to traditional strategies in enclosed, high-risk environments.

Taken together, the results from both controlled laboratory studies and from feasibility and performance validation of air sterilization studies in a nursing home setting consistently demonstrate that CHAST technology is highly effective in eliminating a broad spectrum of airborne biological contaminants, including bacteria, viruses, and resilient fungal spores. These findings validate CHAST as a powerful air sterilization platform capable of functioning reliably across variable environmental conditions. However, an important limitation of this feasibility and performance validation field study is that it was not designed to directly measure HAI incidence or clinical outcomes following CHAST deployment. Despite this limitation, the translational potential of technology is substantial. In high-risk healthcare environments, such as ORs, transplant suites, and LTCFs, where vulnerable patient populations face heightened risk from airborne pathogens, CHAST could offer a transformative infection control solution. This inference is supported by prior research in nursing homes using conventional air purification systems, which showed modest reductions in HAIs linked to airborne and surface pathogen exposure [60]. Given CHAST’s superior pathogen inactivation efficacy (>99.9999% under lab conditions and in field testing and >96% in fungal and mold spores), it is reasonable to anticipate that its clinical deployment could yield even greater benefits, including lower infection rates, reduced length of stay, and decreased hospital costs. Particularly in surgical settings, where environmental sterility is critical, the use of CHAST may contribute to a measurable reduction in surgical site infections (SSIs). To substantiate these anticipated clinical impacts, a prospective research study is currently underway at the same facility to assess CHAST’s effect on infection metrics and patient outcomes. These forthcoming data will be essential in guiding broader adoption and informing regulatory and reimbursement pathways. Beyond its clinical promise, CHAST also offers substantial economic and operational advantages. The main economic advantage by improving IAQ in nursing homes with CHAST system reduces airborne illness transmission, lowering staff absenteeism and outbreak-related costs. Combined with increased Center for Medicare (CMS) reimbursement tied to enhanced safety and infection control, this investment strengthens long-term economic stability. By improving IAQ and reducing microbial exposure for both patients and healthcare workers, the system may help institutions enhance quality-of-care indicators, limit exposure to CMS penalties for hospital-acquired conditions and optimize resource utilization. In summary, CHAST represents a robust, scalable, and clinically relevant innovation in air sterilization, with the potential to significantly elevate infection control practices in healthcare and long-term care environments.

## 5. Conclusions

This study demonstrates that CHAST air sterilization technology offers highly effective, autonomous elimination of airborne pathogens in both laboratory and feasibility study in nursing home healthcare environments. In controlled settings, CHAST achieved >99.9999% inactivation of resilient bacterial spores and viral surrogates. Feasibility field testing in a nursing home confirmed consistent performance, with up to 93% reduction in airborne fungal spores and no viable bacterial growth detected in treated air. These findings highlight CHAST’s capacity to deliver continuous, high-throughput sterilization even under variable, real-life conditions.

While this initial deployment did not assess HAI rates, the marked reduction in airborne bioburden strongly supports the system’s potential to reduce environmental transmission and surface contamination. Ongoing prospective study will evaluate its impact on infection outcomes. As healthcare facilities prioritize air quality and HAI reduction, CHAST offers a scalable, infrastructure-compatible solution with significant implications for long-term care facilities, surgical environments, and other high-risk clinical settings.

## Figures and Tables

**Figure 1 microorganisms-13-02299-f001:**
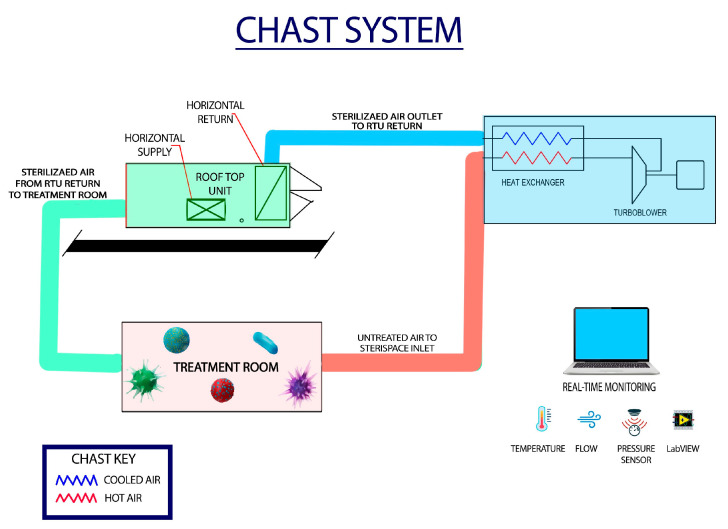
**Schematic representation of the Compressive Heating Air Sterilization Technology (CHAST) system.** Ambient air enters the unit via an intake blower and flows through the untreated intake duct (red), where it undergoes rapid compression and thermal elevation to sterilization temperatures (~240 °C). The heated, sterilized air then passes through a heat exchanger, where it is cooled to near-ambient levels before entering the discharge duct (blue). The cooled sterile air is released into the room through a horizontal return vent and distributed via the Sterilized Air Return Duct (green). Real-time monitoring of temperature, flow rate, and pressure is continuously conducted through an integrated LabVIEW interface to ensure operational stability and reproducibility. Color coding indicates airflow states: Red = intake air, Blue = heated air, Green = sterile air.

**Figure 2 microorganisms-13-02299-f002:**
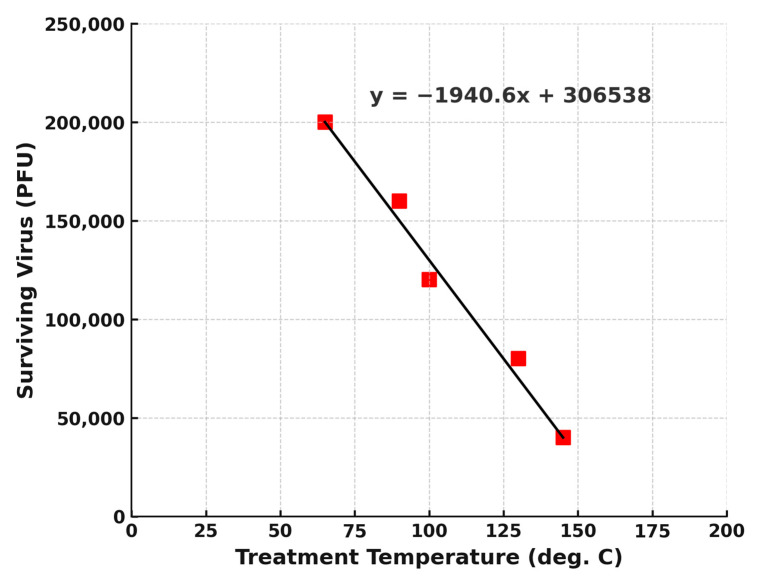
Temperature-Dependent Inactivation of MS2 Bacteriophage Using CHAST Technology. To evaluate the thermal sensitivity of viral inactivation via CHAST, aerosolized MS2 bacteriophage was subjected to treatment temperatures of 143 °C, 103 °C, 91 °C, and 64.5 °C. At each temperature setting, continuous aerosolized samples were collected over a 30-min test interval using an SKC BioSampler. Samples were subsequently cultured overnight, and viral inactivation was quantified the following day. Complete inactivation of MS2 was determined through extrapolation of viral titers to a theoretical zero point. Based on the operating flow rate of the CHAST unit, the calculated residence time of viral particles within the treatment zone was approximately 0.15 s, indicating rapid and effective thermal inactivation under the tested conditions.

**Figure 3 microorganisms-13-02299-f003:**
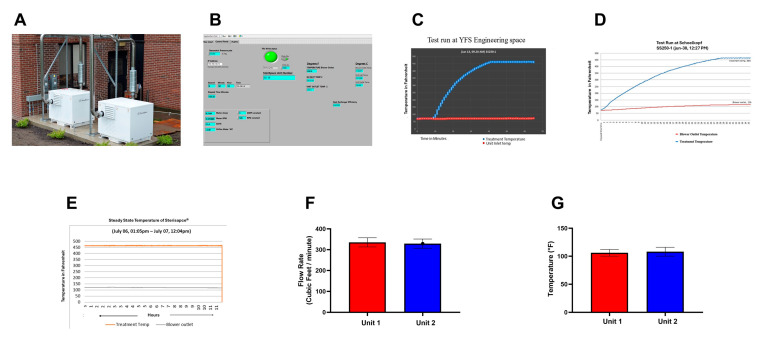
Operational Assessment of CHAST Devices Deployed at Schoellkopf Health Center Under the Clean Air Initiative. As part of a collaborative pilot implementation with Rethink WNY and the City of Niagara Falls, two CHAST units with 300 CFM capacity were installed in the recreational space of the extended care facility at Niagara Falls Memorial Medical Center (NFMMC) to assess real-world operational performance. (**A**) Photograph showing the physical installation of a CHAST device within the Schoellkopf Health Center. (**B**) Screenshot of the custom LabVIEW interface used for real-time monitoring and data acquisition of critical operational parameters. (**C**) Representative time-series plot demonstrating inlet air temperature and the time required to reach the optimal treatment temperature of 240 °C. (**D**) Correlation plot illustrating the relationship between CHAST internal treatment temperature and outlet air temperature during ramp-up. (**E**) Longitudinal monitoring of temperature stability, displaying the steady-state maintenance of treatment and outlet temperatures over an 11-h operational period. (**F**) Quantitative analysis of the mean ± standard deviation (SD) of airflow rates (in CFM) at the outlet of both units, as measured using calibrated handheld flowmeters. (**G**) Mean ± SD of outlet air temperatures recorded inside the facility using handheld infrared thermometers, validating temperature consistency across units.

**Table 1 microorganisms-13-02299-t001:** This table summarizes microbial inactivation efficacy of CHAST prototypes across various development stages. Despite differences in airflow rate, heat exchanger configuration, and system scale, all prototypes demonstrated ≥ 99.9% reduction, with most achieving ≥ 6 log (99.9999%) reduction against aerosolized bacterial and viral agents. These results confirm CHAST’s reproducible high-temperature sterilization capacity across diverse prototype designs.

Test Objective(s)	Temperature (°C)	Challenge Organism(s)	Log or % Kill Rate	Representative Prototype Configurations
**Kill efficacy against bacteria**	240–247	*Bacillus globigii* (Bg), *Bacillus stearothermophilus* (Bst)	>3–6 log (99.9–99.9999%)	Early prototypes (University at Buffalo, DoD GD450/GD540); all configurations achieved ≥99.9% kill
**Kill efficacy against bacteria and virus**	240	*Bacillus globigii* (Bg), *Bacillus thuringiensis* (Bt), *E. coli*, MS2 bacteriophage	>6 log (99.9999%)	DoD HF-408 prototypes (tri-lobe blower, counterflow heat exchanger); efficacy confirmed with mixed bacterial and viral challenge
**Scaled airflow/system optimization**	220–240	*Bacillus globigii* (Bg)	>6 log (99.9999%)	Optimized HF-408 prototypes (alpha/optimized flow configurations) and scaled 5000 CFM centrifugal system
**Maintenance of kill efficacy (long-duration, scaled system)**	240	*Bacillus globigii* (Bg)	>7 log (99.99999%)	5000 CFM centrifugal counterflow system (DoD SBIR program)

**Table 2 microorganisms-13-02299-t002:** Air Sampling Results and Microbial Identification (29 June 2023).

Sampling Location	Bacteria (CFU/m^3^)	Fungi (CFU/m^3^)	Fungal Species (n)	Bacterial Species (n)
**Sitting Area**	OVG	OVG	OVG	OVG
**Front Exit**	35	NGP	-	*Bacillus* sp. (5)
**10’ South of Front Exit**	57	7	*Auerobasidium* sp. (1)	*Bacillus* sp. (3), *Streptococcus* sp. (2), *Actinomycetes* (1), *Gram Neg Bacteria* (1), *Rhodococcus* sp (1)
**20’ South of Front Exit**	21	14	*Cladosporium* sp. (2)	*Bacillus* sp. (2), *Streptococcus* sp. (1)
**Sink**	57	28	*Cladosporium* sp. (4)	*Bacillus* sp. (3), *Staphylococcus* sp. (1), *Streptococcus* sp (1), *Actinomycetes* (2)
**Kitchenette**	7	35	*Cladosporium* sp. (5)	*Bacillus* sp. (5)
**Average Concentration**	35	17		

**Legend Table 2:** This table summarizes the environmental air quality assessment conducted on 29 June 2023, as part of a pre-intervention evaluation of microbial bioburden in a long-term care resident room prior to CHAST activation. Air samples were collected from six predefined locations, targeting high-traffic and transitional areas including the sitting area, front exit, and kitchenette, using an Andersen Airborne Impaction Sampler following NIOSH Method 0800 guidelines. Each air sample was collected over a 5-min interval at a calibrated flow rate of 28.3 L/min (141.5 L total volume). Culture media included 2% malt extract agar (MEA) for fungal identification and trypticase soy agar (TSA) for bacterial growth. Samples were incubated at 25 ± 1 °C for up to seven days. Colony-forming units per cubic meter of air (CFU/m^3^) were calculated for both bacterial and fungal isolates. Species-level identification was performed using macroscopic and microscopic methods. “OVG” denotes overgrowth, where excessive microbial proliferation precluded accurate enumeration. “NGP” refers to “No Growth Promoted” during the incubation period. The average CFU/m^3^ values presented in the final row represent arithmetic means across all locations with valid counts. This baseline data provides a critical benchmark for evaluating the effectiveness of CHAST-mediated air sterilization in reducing airborne microbial loads in healthcare environments.

**Table 3 microorganisms-13-02299-t003:** Baseline Air Sample Analysis.

Collection Date	Sample Location	Sample ID	Volume (L)	Count	Bacterial CFU/m^3^	Bacterial Species Identified	Fungal Count	Fungal CFU/m^3^	Fungal Species Identified
07/20/2023	Supply 2—at outlet	1	141.5	NGP	-	-	1	7	*Aspergillus fumigatus* (1)
07/20/2023	Supply 2–6” from outlet	2	141.5	1	7	*Bacillus* sp. (1)	4	28	*Aspergillus versicolor* (1), *Aspergillus niger* (1), *Alternaria alternata* (2)
07/20/2023	Return 2–6” below diffuser	3	141.5	1	7	*Actinomycete* (1)	7	50	*Acremonium* (3), *Aspergillus versicolor* (1), *Alternaria alternata* (4)
	Mean			<1	<7		4	28	

**Legend Table 3:** This table presents the results of post-treatment air sampling conducted on 20 July 2023, following a 72-h deployment of CHAST (Compressive Heating Air Sterilization Technology) units in a resident room within a long-term care facility. Sampling was performed in accordance with NIOSH Method 0800 using impaction-based collection at a standardized flow rate of 28.3 L/min for five minutes (total volume 141.5 L). Air samples were obtained from three locations representing critical airflow points: the CHAST-treated air outlet, a zone six inches from the outlet, and the return duct inlet located six inches below the ceiling diffuser. Collected samples were cultured on trypticase soy agar (TSA) for bacterial detection and 2% malt extract agar (MEA) for fungal growth, followed by incubation at 25 ± 1 °C for seven days. Colony counts were translated to CFU/m^3^. Identified microbial taxa were confirmed via macroscopic and microscopic characterization. “NGP” indicates No Growth Promoted on culture media, while calculated mean values for bacterial and fungal concentrations reflect arithmetic averages across all tested locations. These data support CHAST’s efficacy in reducing airborne bioburden at treated air discharge points compared to untreated return zones, with notable suppression of fungal spores and near-complete bacterial elimination at the outlet.

**Table 4 microorganisms-13-02299-t004:** Repeat Sample Collection and Analysis (9 August 2023).

Collection Date	Sample Location	Sample ID	Volume (L)	Count	CFU/m^3^	Status
8/9/23	Supply 2–at outlet	1	141.5	0	<7	No Growth Promoted (NGP)
8/9/23	Return–at outlet	1	141.5	0	<7	No Growth Promoted (NGP)

**Legend Table 4:** This table presents post-treatment air sampling results obtained on 9 August 2023, during evaluation of the CHAST (Compressive Heating Air Sterilization Technology) system operating in a long-term care facility resident room. Following a continuous sterilization period of at least 72 h, air samples were collected at two critical locations: the treated air outlet (‘Supply 2–at outlet’) and the return air inlet (‘Return–at outlet’). Sampling was performed using an impaction sampler following NIOSH Method 0800 protocols. A total air volume of 141.5 L per sample was drawn at a calibrated flow rate of 28.3 L/min over a 5-min period. Collected samples were cultured on general-purpose nutrient media and incubated at 25 ± 1 °C for seven days. Microbial colonies were assessed at regular intervals using macroscopic observation. Both locations demonstrated complete absence of viable microbial growth, indicated as ‘No Growth Promoted’ (NGP). Corresponding colony-forming unit (CFU) values were reported as ‘<7 CFU/m^3^,’ representing the detection threshold for the method under these conditions. These results validate the CHAST system’s high-level sterilization capability in actively occupied healthcare environments.

## Data Availability

The original contributions presented in this study are included in the article/Appendix A. Further inquiries can be directed to the corresponding author.

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
