# Peer review of "Elimination of Airborne Microorganisms Using Compressive Heating Air Sterilization Technology (CHAST): Laboratory and Nursing Home Setting"

_microorganisms, 2025, doi:10.3390/microorganisms13102299_

Round 1
Reviewer 1 Report
Comments and Suggestions for Authors
Abstract:
The provided abstract does seem to read more like a promotional advertisement than a scientific abstract. It needs to be improve significantly.
The methodology lacks any technical or engineering data on the equipment's operation; an engineering diagram of the system design needs to be included. There are many questions about its operation, for example, if the air is heated to 240 °C, how is it subsequently cooled? The authors need to provide more detailed information about the proposed equipment in their methodology. Furthermore, the methodology used to monitor the equipment’s performance does not adhere to any national or international standards for conducting such analyses, which means the results obtained once again lack the scientific rigor required for studies of this nature.
I insist that the authors must provide those details about their technology since the main innovative characteristic of their work is its use.
Despite the experimental effort made with the proposed technology, if there are no details about the operation of the equipment, it is not possible to analyze the veracity of the data provided in the results.
It is necessary to include an energy and cost balance of the proposed technology compared to similar technologies available in the market.
Author Response
Reply in the uploaded Rebuttal Document "Reviewer comments1".

Reviewer 2 Report
Comments and Suggestions for Authors
This study evaluated the sterilization efficacy of CHAST (Compressed Heating Utetheisa Kong Gas Sterilization Technology) in both laboratory and nursing home environments, addressing the gap in the application of high-temperature sterilization technology in medical settings. The results demonstrated that CHAST can effectively inactivate bacteria, viruses, and Phoxinus phoxinus subsp. phoxinus spores (>99.9999%), highlighting its significant value in improving air quality in high-risk environments such as nursing homes.However, I think this article needs to be revised before it can be published. The specific opinions are as follows.
1.Please further clarify whether the source of residual microorganisms in the nursing home tests (e.g., 7 CFU/m³ at the exit) is related to system limitations or environmental recontamination.
2.In Figure 2, it is recommended to keep the border sizes of Figure 2(F)and(G)consistent for better aesthetics.
3.The size of images annotations should be kept as consistent as possible.
4.The energy consumption of high-temperature sterilization and its compatibility with the HAVC system were not specified. Since it is intended for nursing homes, is this system suitable for practical implementation?
5.In Table1,Multiple prototype results are similar and can be briefly described.
Author Response
Reply in uploaded Reviewer 2 comments uploaded

Reviewer 3 Report
Comments and Suggestions for Authors
This study evaluates the Compressive Heating Air Sterilization Technology (CHAST), a system that sterilizes air by using extreme heat (up to 240 °C) without filters or chemical agents. The goal is to reduce the spread of airborne microorganisms—bacteria, viruses, and molds—in high-risk environments such as nursing homes, where healthcare-associated infections (HAIs) are frequent and have severe consequences for residents’ health.
The research was carried out in two phases: laboratory testing and deployment in a nursing home in Niagara Falls, NY. In the lab, CHAST achieved over 99.9999% (>6 log) inactivation of highly resistant microorganisms, including bacterial spores (Bacillus globigii, B. stearothermophilus, B. thuringiensis), vegetative bacteria (E. coli), and the MS2 bacteriophage, used as a viral surrogate. The study also investigated the minimum temperature needed for viral inactivation, estimated at ~170 °C, with progressive reduction in viral load even at lower levels.
In the real-world nursing home setting, two CHAST units operated continuously for 72 hours, treating and recirculating air in a common room. Pre-treatment air sampling revealed widespread contamination with environmental and commensal bacteria (Bacillus spp., Staphylococcus spp., Streptococcus spp.) and common molds (Cladosporium, Penicillium, Trichophyton). After treatment, the air exiting the CHAST units contained no viable microorganisms, and there was a 93% reduction in fungal spores compared with untreated return air. Minor traces of contamination (7 CFU/m³) detected near air outlets were attributed to environmental re-entry rather than system failure.
The paper highlights CHAST’s advantages over conventional technologies such as HEPA filtration and UVGI, including continuous operation, the ability to inactivate even highly resistant pathogens, and removal of volatile organic compounds (VOCs). However, it acknowledges a key limitation: the study did not directly measure reductions in HAI incidence, leaving that to prospective clinical trials already in progress.
Lack of direct clinical outcome data
While the results show very high microbiological efficacy (>99.9999% in the lab and no downstream growth in the field), the study does not measure the actual reduction in healthcare-associated infections among nursing home residents, focusing solely on air microbiology (line 491–496). Without clinical data, the real-world impact on morbidity and mortality remains speculative.
Potential for uncontrolled re-contamination
The detection of Bacillus spp. and Actinomycetes near air outlets (7 CFU/m³) indicates that system performance may be affected by environmental intrusion (line 303–307, 449–455). In critical applications, CHAST should be paired with environmental control protocols to maintain “cleanroom-level” standards.
No direct quantitative comparison with other technologies
Although the paper discusses limitations of HEPA and UVGI (line 399–405), it does not present a side-by-side comparison of CHAST and these systems under equivalent conditions. Controlled comparative testing would provide stronger evidence of CHAST’s added value.
Cost and implementation feasibility not addressed
The paper does not provide details on purchase, maintenance, and energy consumption costs, or on integration into existing HVAC systems. Although potential economic benefits are mentioned (line 508–511), no quantitative cost–benefit analysis is presented.
Need for assessment of exposure times and occupant comfort
The study focuses on microbiological efficacy at 240 °C but does not explore how operational variables (airflow, perceived outlet temperature, noise) might affect resident comfort and energy use (line 243–252). Furthermore, while flexibility to operate at lower temperatures is mentioned (line 217–230), this was not validated in real-world settings.
Author Response
Reply in uploaded Reviewer 3 comments.

Round 2
Reviewer 1 Report
Comments and Suggestions for Authors
The article titled "Elimination of Airborne Microorganisms Using Compressive Heating Air Sterilization Technology (CHAST): Laboratory and Nursing Home Setting" has been significantly improved by the authors and it is now possible for its publication in the journal of Microorganisms.